# Light Increases Astaxanthin Absorbance in Acetone Solution through Isomerization Reactions

**DOI:** 10.3390/molecules28020847

**Published:** 2023-01-14

**Authors:** Oleksandr Virchenko, Tryggvi Stefánsson

**Affiliations:** Algalif Iceland ehf., Bogatrod 10, 262 Reykjanesbaer, Iceland

**Keywords:** astaxanthin isomers, *Haematococcus pluvialis* oleoresin, astaxanthin quantitative analysis, astaxanthin light sensitivity

## Abstract

Astaxanthin quantitative analysis is prone to high variability between laboratories. This study aimed to assess the effect of light on the spectrometric and high-performance liquid chromatography (HPLC) measurements of astaxanthin. The experiment was performed on four *Haematococcus pluvialis*-derived astaxanthin-rich oleoresin samples with different carotenoid matrices that were analyzed by UV/Vis spectrometry and HPLC according to the United States Pharmacopoeia (USP) monograph. Each sample was dissolved in acetone in three types of flasks: amber glass wrapped with aluminium foil, uncovered amber glass, and transparent glass. Thus, the acetone solutions were either in light-proof flasks or exposed to ambient light. The measurements were taken within four hours (spectrometry) or three hours (HPLC) from the moment of oleoresin dissolution in acetone to investigate the dynamics of changes in the recorded values. The results confirm the logarithmic growth of astaxanthin absorbance by 8–11% (UV/Vis) and 7–17% (HPLC) after 3 h of light exposure. The changes were different in the samples with different carotenoid matrices; for instance, light had the least effect on the USP reference standard sample. The increase in absorbance was accompanied with the change of isomeric distribution, namely a reduction of 13Z and an increase of All-E and 9Z astaxanthin. The greater HPLC values’ elevation was related not only to the increase of astaxanthin absorbance, but also to light-dependent degradation of internal standard apocarotenal. The findings confirm a poor robustness of the conventional analytical procedure for astaxanthin quantitation and a necessity for method revision and harmonization to improve its reproducibility.

## 1. Introduction

Astaxanthin (3,3′-dihydroxy-β-carotene-4,4′-dione) is a carotenoid pigment, for which multifunctional health benefits and safety are reported widely [1,2]. While synthetic preparations dominate in the feed market, a natural astaxanthin is being exclusively used in pharmaceutical, cosmetic, and food industries [3]. Only a few sources of natural astaxanthin are known, among which a fresh water green algae *Haematococcus pluvialis* is the most preferred choice [4].

Astaxanthin oleoresin derived from the algal biomass is represented by the mix of all-E-isomer and 9Z- and 13Z-isomers with all-E-form being a main component [5]. There are multiple approaches to the quantitation of astaxanthin in samples by high-performance liquid chromatography (HPLC) using isocratic or gradient mobile phases in either normal-phase or reversed-phase mode [6,7,8,9,10]. The availability of different analytical methods increases the measurement uncertainty; thus, the same sample can be measured as containing ±20% of astaxanthin, which understandably disrupts operations within the astaxanthin industry and the same astaxanthin level stated on a label varies significantly between producers. 

The commonly used validated analytical procedure described in the United States Pharmacopoeia (USP) monograph (USP 42) suggests using a reversed-phase HPLC with a three-component gradient mobile phase (81% methanol, 15% t-butylmethylether, 4% phosphoric acid (1% aqueous)) for astaxanthin quantification [11]. The procedure also defines different response factors for astaxanthin Z isomers—1.3 for 13Z and 1.1 for 9Z—at wavelength 474 nm. However, the verification of this method performed in the quality department showed low reproducibility of the method. On the one hand, low reproducibility is related to the presence of different astaxanthin isomers (13Z, 9Z, All-E, 15Z) and their variable relative ratio in the samples, as well as other carotenoids, which absorb light at wavelength 474 nm [12]. Thus, a HPLC chromatogram contains many carotenoid peaks, which are located close to each other and can be partially merged depending on the HPLC system settings. This decreases a resolution between adjacent peaks and impairs astaxanthin quantification. On the other hand, the complexity of the analytical procedure—enzymatic hydrolysis of esters, extraction with petroleum ether—adds to the method uncertainty. However, laboratory trials recently demonstrated that light and heat are the main factors affecting astaxanthin analysis, with light being the most important. It was found that the same oleoresin sample dissolved in acetone in a transparent glass flask, therefore being exposed to ambient light, possesses higher UV/Vis absorbance than the sample protected from light by covering the flask with aluminium foil. Interestingly, this change happens only in an acetone solution of astaxanthin since direct illumination of oleoresin does not impact its absorbance. Moreover, it was noticed that the level of astaxanthin isomers is not the same within repeated measurements of the same sample and might be a key to understanding the observed light-induced increase of astaxanthin level. The observations are supported by scientific evidence about astaxanthin isomerization reactions [5,13,14]. The conversion of trans-astaxanthin to cis-astaxanthin under heat impact [13,14,15], as well as isomerization reactions of other carotenoids [16,17] have been demonstrated. However, the information regarding the light-induced increase of astaxanthin absorbance and transformation of its isomers in acetone solution is lacking.

Therefore, the aim of this study is to evaluate all effects of light on astaxanthin analysis, both spectrometric and HPLC, namely the growth of astaxanthin absorbance in an acetone solution and isomerization events depending on the carotenoid composition of samples, thus, collecting data for further revision, improvement, and eventually, harmonization of the astaxanthin analytical procedure within the industry.

## 2. Results

### 2.1. Spectrometric Analysis of Astaxanthin

The samples of three batches of Astalif 10 (astaxanthin-rich oleoresin) and the reference USP astaxanthin standard were dissolved in acetone and diluted in three types of volumetric flasks: clear glass, amber glass, and amber glass covered with aluminium foil, and then exposed to light for 4 h. Spectrometric analysis showed the increase of UV/Vis absorbance of astaxanthin in an acetone solution in all flasks that were not fully protected from light—both amber and clear glass flasks (Figure 1b,c). The effect was more pronounced in the latter: the absorbance increased by 5% in transparent flasks after one hour of staying under light compared to 2% in amber glass flasks. After two hours, the corresponding increase was 7% against 4% (Figure 1b,c). The UV/Vis growth is well described by the logistic 3P model:(1)$$\mathrm{Astaxanthin}\;\%\left(\mathrm{time}\right)=\frac{c}{1+e^{\left(-a\times \left(\mathrm{time}-b\right)\right)}}$$
where a—growth rate, b—inflection point, c—asymptote.

The parameters of the logistic curve describing the UV/Vis growth are presented in Table 1.

The deviation of absorbance in the control samples in an amber flask wrapped with aluminium foil did not exceed 3%, which is within uncertainty of the method; slight increase of values, however, can be explained by acetone evaporation during the experiment (Figure 1a). 

The UV/Vis absorbance of the samples from all lots increased after light exposure, but the growth pattern was not identical between the samples. A rate of growth was much higher in the Astalif 10 samples (O10-200, O10-205, O10-208) than in the astaxanthin reference standard and reached 10% in O10-200 and O10-205 (Figure 2). The growth curve parameters were also different between the samples from different lots (Table 2). It must be noted that the logistic curve was not statistically significant for the reference standard dissolved in amber glass flasks. Also, according to the prediction model, the reference standard astaxanthin level after dissolution in a transparent flask should reach 11% after 6 h; however, it has not been verified experimentally.

### 2.2. HPLC Analysis of Astaxanthin

The HPLC analysis also showed the same patterns of absorbance growth after light exposure (Figure 3). The logistic 3P model fits the Astalif 10 samples’ growth curves well, however, it does not fit the reference standard, in which absorbance increased much more slowly. Thus, more data are needed to estimate the astaxanthin growth pattern of the standard (Table 3). Moreover, the curves were not equal even between the Astalif 10 samples and the astaxanthin level elevation ranged from 12 to 17% after three hours of light exposure (Figure 3). Thus, the Astalif 10 samples were 5–10% higher than the USP standard after 3 h under ambient light despite all four samples being analyzed as 10% by the “no-light” approach. 

The HPLC results were not fully aligned with the UV analysis (Figure 2 and Figure 3). While the UV/Vis raise of O10-200 was the lowest among all the Astalif 10 samples, its HPLC values increased the most. Moreover, the HPLC total absorbance increased more significantly than the UV/Vis absorbance, and this can be explained by not only the increase of the astaxanthin level itself, but also the light-induced degradation of the internal standard apocarotenal. Since the ratio of the analyte signal/internal standard signal is compared between samples, the lower level of the apocarotenal is, the higher final value of total absorbance is recorded. Indeed, the apocarotenal peak area reduced by at least 4% in Astalif 10 after 3 h (Figure 4d). 

The HPLC total absorbance increase was accompanied by significant changes in the HPLC peak areas of astaxanthin isomers, as well as other carotenoids (Figure 4, Figure 5, Figure 6 and Figure 7). There was a decrease of the 13Z astaxanthin peak area by more than double after 3 h of light exposure of an acetone solution in all the Astalif 10 samples (O10-200, O10-205, O10-208) (Figure 4a). 

All-E astaxanthin increased by more than 20% in all the Astalif 10 samples (Figure 4b). The 9Z astaxanthin increased by more than 20% in O10-200 and O10-208, and more than 10% in O10-205 (Figure 4c). However, the changes in the reference standard astaxanthin isomers were less evident. After 3 h, 13Z astaxanthin dropped by only 19%, and All-E and 9Z astaxanthin increased by 7% and 6%, respectively (Figure 4). Apocarotenal decreased in standard as well, but the pattern of the change differs from Astalif 10 (Figure 4 and Figure 6). Importantly, the initial (“no-light”) concentration of apocarotenal in the standard was lower by more than 4% than in Astalif 10; however, it is expected to be at the same level (Figure 6). This potentially results in an overestimation of the standard total absorbance if tested in “no-light” conditions.

It is noted that the patterns of the change in the astaxanthin isomers and apocarotenal level in the USP standard are completely different from those of the Astalif 10 samples (Figure 4), which might indicate the influence of the carotenoid matrix on the light-induced isomerization processes. Interestingly, an initial (“no-light”) isomeric distribution (relative ratio of astaxanthin peaks) in all oleoresin samples was similar (Figure 5). The only significant difference was the 9Z level in the USP standard, which accounts for 9% of all astaxanthin, while it was around 13% in the Astalif 10 samples or 30% lower (Figure 5). However, after light exposure, the standard’s isomeric distribution differs significantly from other oleoresin samples: the 13Z astaxanthin relative level was two times higher, 9Z was more than 1.5 times lower, and All-E was 4% lower in the USP standard than in the samples of Astalif 10 (Figure 5).

Other carotenoids underwent significant changes as well, which is illustrated in Figure 8. The decrease of the carotenoid peaks with retention time 7.4 and 7.7 min and the increase of the peak at 12.2 min (Figure 8) were noted. However, due to these multidirectional reactions, the total area of these peaks tends to decrease slightly: a 2% drop was recorded in the Astalif 10 samples, but there was no difference in the reference standard (Figure 7).

Typical chromatograms of the USP standard and the sample of O10-205 measured with the “no-light” approach and after 3-h light exposure are presented in Figure 8. The above-mentioned changes, namely, a decrease of 13Z astaxanthin and apocarotenal peaks, an increase of All-E and 9Z astaxanthin peaks, as well as a decrease of the carotenoid peaks at 7.4 min and an increase of the carotenoid peak at 12.2 min, are noticeable in Figure 8d.

## 3. Discussion

The study findings demonstrate the intrinsic behaviour of astaxanthin isomers in an acetone solution and confirm that its exposure to light changes the UV absorbance of the solution quickly and significantly, which results in the increase of the recorded astaxanthin level. Apparently, this change of total absorbance is related to the isomerization events in an acetone solution caused by light, namely, the transformation of 13Z astaxanthin to All-E and 9Z astaxanthin. Indeed, the capacity of 13Z astaxanthin to absorb light is lower compared to All-E astaxanthin, meaning that the same amount of 13Z astaxanthin has lower absorbance than All-E. Thus, the sample which contains a higher level of 13Z astaxanthin is tested spectrometrically as having a lower astaxanthin percentage even if both samples have the same quantity of astaxanthin molecules. A similar but lower difference exists between 9Z and All-E astaxanthin. The HPLC method accounts for this difference by multiplying the HPLC peak areas of 13Z and 9Z astaxanthin by response factors 1.3 and 1.1 respectively [11]. Therefore, because of the presence of different astaxanthin isomers with different absorbance rates and in different ratios in oleoresin, the UV method cannot be as precise as the HPLC method. Additionally, the isomerization reaction of astaxanthin under light adds a significant amount of overall uncertainty in the UV method since the astaxanthin level might fluctuate to up to 10% (Figure 1). Indeed, there is no accurate UV/Vis approach here. Measurement under “no light” gives lower numbers because of the high level of 13Z astaxanthin. The light approach (analysis after at least 2 h of light exposure) is closer to the true value since at least 80% of the isomers are All-E astaxanthin and the calibration curve is made using the All-E astaxanthin standard. At the same time, the “no-light” UV/Vis approach is faster and gives more stable and reproducible numbers when the carotenoid matrix is the same. The carotenoid matrix influence on the analysis is clearly visible in Figure 2, where the light-induced growth of astaxanthin level is different between the oleoresin samples.

The presence of astaxanthin isomers with different absorbance rates makes an HPLC method preferable for astaxanthin quantification, but at the same time, it is also significantly affected by light exposure of the acetone solution. Despite similar transformation events in all oleoresin samples, namely a decrease of the 13Z astaxanthin and an increase of the All-E and 9Z astaxanthin HPLC peak area, the light-induced changes are not equal in Astalif 10 samples and the reference standard. Since the standard absorbance increase speed is lower than in Astalif 10 samples, their quantification against the standard shows higher values than in the “no-light” approach (Figure 3). For instance, the absolute absorbance increase in the oleoresin sample O10-200 was 17%, and the standard increased by 7% at the same time, thus, the percentage of O10-200 appears to be higher by approx. 10% after 3-h light exposure.

The differences in the carotenoid matrices must be the key factor of the isomerization reactions’ variability in the oleoresin samples. However, the only pronounced feature of the reference standard is significantly lower 9Z astaxanthin. It is not clear how the makeup of the carotenoid matrix makes the standard more resilient to light exposure.

It was also shown that the drop (300–400 absorbance units) in 13Z astaxanthin after light exposure does not align with the total increase (500–700 absorbance units) of All-E and 9Z astaxanthin in the Astalif 10 samples, which is almost two times higher. This may suggest that either the response coefficients of cis astaxanthin isomers are not correct, or there are other isomerization reactions in place. Indeed, the USP monograph method requires the use of response factors 1.3 and 1.1 for 13Z and 9Z astaxanthin, respectively, however, other alternative approaches state 1.6 and 1.13 as response factors. Thus, to improve the method, the response factors must be reassessed taking into account light-induced isomerization events. 

It must be noted that while the increase of UV/Vis absorbance was 8–11%, the HPLC absorbance increased by 7–17%. A higher maximum increase of HPLC values is related not only to an increase of the total astaxanthin absorbance, but also to light-induced degradation of apocarotenal. Since total absorbance is calculated by dividing by the apocarotenal absorbance, lower apocarotenal values make total absorbance higher. Surprisingly, the apocarotenal level, which must be very similar in all samples, was, in fact, lower by 4% in the standard (Figure 4). This makes its final values higher and consequently underestimates the Astalif 10 samples. It is unclear why this happens, but a plausible explanation may be the matrix effect on the degradation/isomerization of apocarotenal and/or on the extraction coefficient by petroleum ether. Moreover, the degradation profile of apocarotenal is similar between the Astalif 10 samples and the pure apocarotenal solution, but the curve slope of apocarotenal in the standard is lower (Figure 6). 

There are at least six more HPLC peaks on the chromatograms apart from the three mentioned astaxanthin peaks and apocarotenal. At the same time, the astaxanthin oleoresin sample’s carotenoid matrix includes other astaxanthin isomers (15Z, 9,9′-di-cis, 9,13′-di-cis, etc.) and other carotenoids, like β-carotene, lutein, zeaxanthin, and canthaxanthin [12,18,19]. Their total level, however, does not exceed 10% in most cases. A slight decrease by 2–4% of these carotenoids’ absorbance in Astalif 10 samples after light exposure was established, thus, the samples appeared to have a higher astaxanthin purity (95% vs. 92%) if analyzed in ambient light. Individual carotenoid peaks changed even more drastically: peaks at 7.3–8.8 min reduced significantly, but peaks at 12.2–12.8 increased instead. Chemical reactions and their constants associated with these changes are yet to be established. 

The study’s results explain the causes of the UV/Vis and HPLC method uncertainty and point out at flaws in the USP method. It also appears unclear which of the HPLC approaches yields results closer to a true value. Protecting samples from light reduces variability and makes it possible to record the true isomeric distribution, which is not affected by light. Therefore, the “no-light” approach is validated and used in the quality control department. At the same time, it increases the chance of underestimation of the oleoresin samples. Moreover, the EU regulation regarding astaxanthin quality establishes the level 13Z isomer as lower than 7% and all-E form as higher than 75% from other isomers [20,21]. Such a level is recorded only by “light” approach since the average 13Z astaxanthin relative level is 18–24%, and all-E is 60–72% from all astaxanthin isomers if measured in “no-light” conditions. Thus, the revision and improvement of an astaxanthin analysis is especially important for the harmonization between companies and regulation of the astaxanthin industry.

Underlying chemical mechanisms associated with light exposure need to be elucidated in further studies. Contrarily, heat-induced astaxanthin isomerization processes have already been published [13,14]. Interestingly, heat provokes opposite isomerization processes—conversion of trans-astaxanthin to cis-astaxanthin mainly to 13Z isomer associated with a drop of total absorbance [22,23]. Whether heating the samples to 37 °C for 45 min for hydrolysis of astaxanthin esters during sample preparation impact the isomerization process is presently unknown. However, only few studies attempted to investigate the light impact on E/Z isomerization [5,24]. Viazau et al. [5] showed that under prolonged illumination, the content of both Z-isomers decreases. However, the study does not mention the conversion of cis-astaxanthin to trans-form instead explained the drop in Z-isomer by oxidation and the formation of epoxy- and apo-products [5]. It must be noted that in our study, the illumination of the acetone solution was performed after hydrolysis of the esters during the HPLC preparation procedure. Contrarily, the esterified astaxanthin solution was exposed to light for the UV/Vis analysis. Esterification might have an impact on cis-trans equilibrium, as it was shown recently that free astaxanthin produces larger amounts of 9Z isomer and monopalmitate esterification results in increased 13Z isomerization instead [25]. However, this study was performed on separate astaxanthin molecules, whereas algal-derived natural oleoresin contains a mixture of carotenoid molecules, thus, additional studies are needed to discover any difference in the light-induced cis to trans-astaxanthin conversion between free and esterified astaxanthin.

Developing a better astaxanthin quantification method requires the identification of all mechanisms of transformation between astaxanthin isomers, a revision of response factors for 13Z and 9Z astaxanthin, and elucidating why different carotenoid matrices behave differently after exposure to light. The conventional USP or similar approaches, however, lead to a high deviation of the measurement, thus, the difference of more than 10% between the laboratories is not surprising.

## 4. Materials and Methods

### 4.1. Selected Astaxanthin Oleoresin Batches

The study was performed on four 10% astaxanthin oleoresin samples with different carotenoid composition: the USP reference standard and three Astalif 10 samples (O10-200, O10-205, O10-208). All oleoresin lots chosen for the study had a different ratio between UV and HPLC, which corresponds to the ratio between the total carotenoid complex and the astaxanthin isomers levels. The Astalif 10 batches also had a different sunflower oil content (Table 4). 

### 4.2. UV/Vis Spectrometric Analysis

All four oleoresin samples were heated up at 70 °C for exactly 10 min. Approximately 23 mg of each sample was weighed out into a glass-weighing boat with built-in funnel and then dissolved in acetone in 100 mL amber glass flasks. The flasks were immediately wrapped with aluminium foil (Figure 9a). Each solution was diluted 10 times into three types of flasks: amber glass wrapped with foil, uncovered amber glass, and clear glass flasks. The solutions were measured spectrometrically at wavelength 478 nm immediately and after 3, 6, 10, 15, 20, 25, 30, 40, 50, 60, 75, 90, 105, 120, 150, 180, and 210 min from dilution. Each time the absorbance was recorded twice. The quantification was calculated against a calibration curve (*y* = 4.727*x* − 0.025, *y—*astaxanthin in the sample (mg), *x*—absorbance) made with a synthetic all-trans astaxanthin standard (AdipoGen^®^, Liestal, Switzerland). The astaxanthin weight was converted into a mass fraction dividing by the sample weight. Each oleoresin’s UV result at 0 min after dilution was converted to 10%. The obtained coefficients were used to convert all the UV values. 

### 4.3. HPLC Analysis

#### 4.3.1. Astaxanthin and Apocarotenal Stock Solutions Preparation

The same samples in 100 mL flasks (the USP standard, O10-200, O10-205, O10-208) were used as stock solutions for the HPLC analysis, which was performed according the USP42 monograph [11]. Approximately 1.8 mg of apocarotenal (internal standard) was weighed and dissolved with acetone in a 50 mL amber glass flask covered with aluminium foil.

#### 4.3.2. Enzymatic Hydrolysis of Astaxanthin Esters

For each oleoresin batch, three Pyrex tubes placed into aluminium pouches for light protection were charged with 2 mL of astaxanthin stock solution, 1 mL of apocarotenal stock solution, and 3 mL of Cholesterol esterase solution (4.5 units/mL of the enzyme in buffer solution Tris-HCl, pH 7.0) each. The reagents were mixed gently by inversion and the tube was placed in a block heater set to 37 °C allowing the reaction to continue for 45 min, gently and slowly inverting the tube every 10 min. The samples were covered with aluminium foil to protect from ambient light while in the block heater (Figure 9b).

#### 4.3.3. Extraction of Hydrolyzed Astaxanthin and Preparation of Samples for HPLC

After hydrolysis was complete, all 12 Pyrex tubes were put back into aluminium pouches, and 1 g of Na_2_SO_4_•10H_2_O and 2 mL of petroleum ether were added. The tubes were mixed in a vortex mixer for 30 s and centrifuged at 1500 rcf for 3 min allowing separation of liquid phases and extraction of astaxanthin into the upper petroleum ether layer. The upper layer was transferred to a new tube containing 1 g of anhydrous sodium sulfate, vortexed, and evaporated under a steam of nitrogen. After drying, the tubes were placed again in aluminium pouches and the material was reconstituted with 3 mL of acetone. The solution then was sonicated and vortexed. The obtained mixture was filtered through a 0.45 µm filter prior to the HPLC analysis. Each HPLC vial was charged with approximately 0.3 mL from each of three Pyrex tubes from the same batch to make up 1 mL.

#### 4.3.4. HPLC Analysis Design

A final analytical solution of all batches, as well as the apocarotenal sample, was measured by HPLC immediately after preparation (“no-light” approach). Then they were exposed to light and after 15, 30, 45, 60, 90, 120, and 180 min of being illuminated by ambient light (“light” approach), and the correspondent aliquots were taken for the HPLC analysis (Table 5).

#### 4.3.5. Astaxanthin Level Calculation

The quantification of all samples was calculated against the first “no-light” value of the USP standard. The initial “no-light” values of the Astalif 10 samples were converted to 10%, and all further HPLC values were reduced correspondently by the obtained conversion coefficient.

The calculation of the astaxanthin level was obtained according to the formula:(2)$$\mathrm{AXT}\%=\frac{{\mathrm{AXT}}_{\mathrm{sample}}\times m_{\mathrm{STD}}\times 10\times {\mathrm{IS}}_{\mathrm{STD}}}{{\mathrm{IS}}_{\mathrm{sample}}\times m_{\mathrm{sample}}\times {\mathrm{AXT}}_{\mathrm{STD}}}$$
where AXT%—the astaxanthin level in the sample; AXT_sample_, AXT_STD_—total astaxanthin absorbance in the sample and standard; m_sample_, m_STD_—the weight of the sample and standard; 10—the astaxanthin level (10%) in the standard; and IS_sample_, IS_STD_—the peak area of apocarotenal (internal standard) in the sample and standard.

The total astaxanthin absorbance was calculated as a sum of the astaxanthin isomers (13Z, All-E, 9Z) peak areas multiplied by response factors:(3)AXT=1.3×13Z+AllE+1.1×9Z
where AXT—total astaxanthin absorbance in the sample or standard; 13Z—the peak area of 13Z astaxanthin; All-E—the peak area of All-E astaxanthin; and 9Z—the peak area of 9Z astaxanthin.

#### 4.3.6. Characteristics of the Chromatographic System

The HPLC analysis was performed with an Agilent 1260 Infinity I HPLC system coupled with a diode array detector (DAD). The HPLC column was a YMC Carotenoid, C30 reversed-phased column, 4.6-mm × 25-cm, 5-µm packing L62. The column was set up at 25 °C with a flow rate of 1 mL/min and an injection volume of 20 µL. The mobile phase included methanol, t-butylmethylether, and 1% aqueous phosphoric acid with a gradient presented in Table 6. The detection of astaxanthin absorbance was performed at 474 nm. The relative retention time was 0.9 for 13Z astaxanthin, 1 for All-E astaxanthin, 1.3 for 9Z astaxanthin, and 1.6 for apocarotenal.

### 4.4. Statistical Analysis

Statistical analysis of the data was performed with JMP Pro^®^ 16 software (JMP Statistical Discovery LLC., Cary, North Carolina).

## 5. Conclusions

Light induces the isomerization process of astaxanthin in an acetone solution, which results in the decrease of 13Z and increase of All-E and 9Z isomers, as well as transformation of other carotenoids. The internal standard apocarotenal degrades/transforms under light impact. These processes elevate the relative level of astaxanthin up to 20%. Thus, light is a significant factor of the astaxanthin quantification method uncertainty. The analytical procedure must be revised and harmonized within the astaxanthin industry.

## Figures and Tables

**Figure 1 molecules-28-00847-f001:**
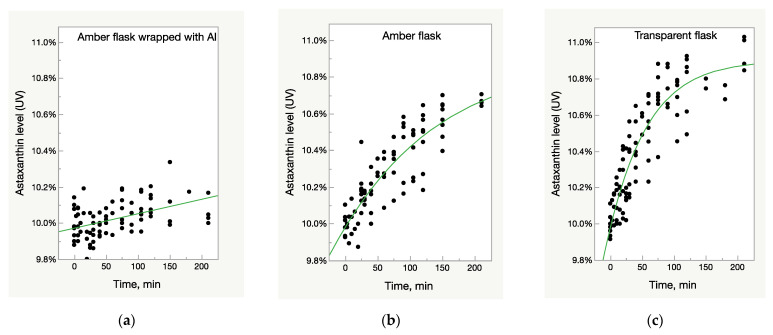
Light-dependent increase of UV/Vis absorbance in oleoresin samples. (**a**) Control samples—oleoresin dissolved in acetone in amber glass flasks wrapped with aluminium foil and thus, fully protected from light impact. (**b**) Samples in amber glass flasks partially protected from light impact. (**c**) Samples in clear glass flasks fully exposed to light impact.

**Figure 2 molecules-28-00847-f002:**
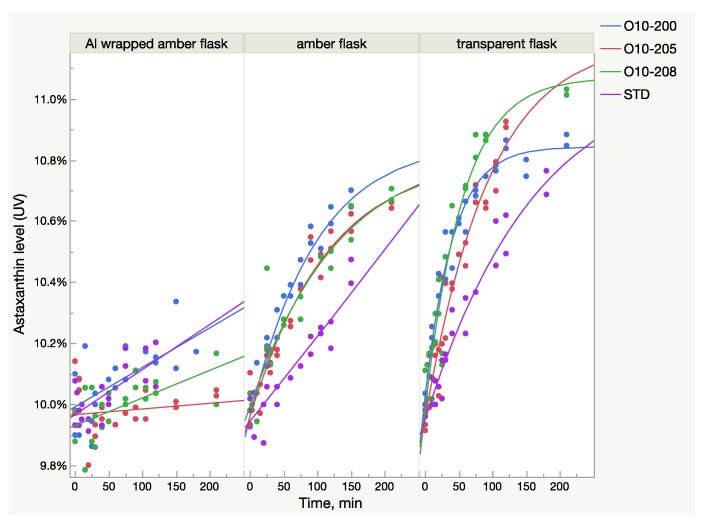
Light-dependent increase of UV/Vis absorbance in oleoresin samples in three types of flasks broken down by sample lots.

**Figure 3 molecules-28-00847-f003:**
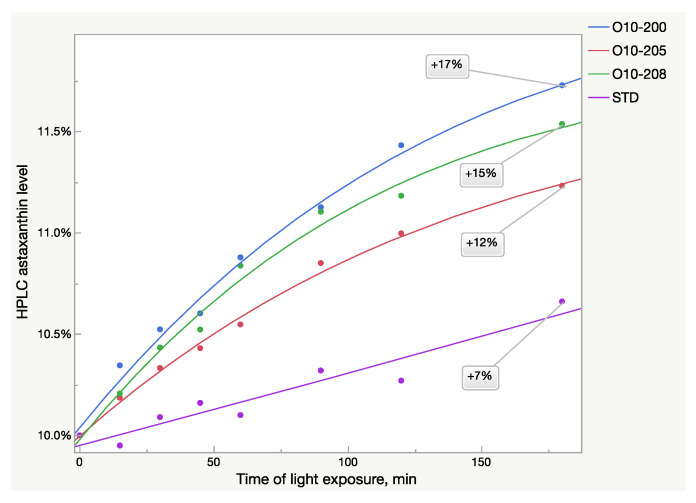
Light-dependent increase of HPLC astaxanthin level in oleoresin samples broken down by sample lots.

**Figure 4 molecules-28-00847-f004:**
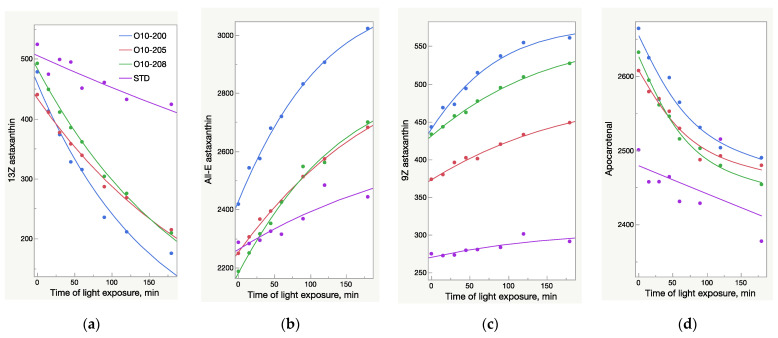
Light-dependent change of astaxanthin isomers and apocarotenal peaks area (mAU×s) in oleoresin samples broken down by sample lots; AU—absorbance units. (**a**) Decrease of 13Z astaxanthin peak area. (**b**) Increase of All-E astaxanthin peak area. (**c**) Increase of 9Z astaxanthin peak area. (**d**) Decrease of apocarotenal peak area.

**Figure 5 molecules-28-00847-f005:**
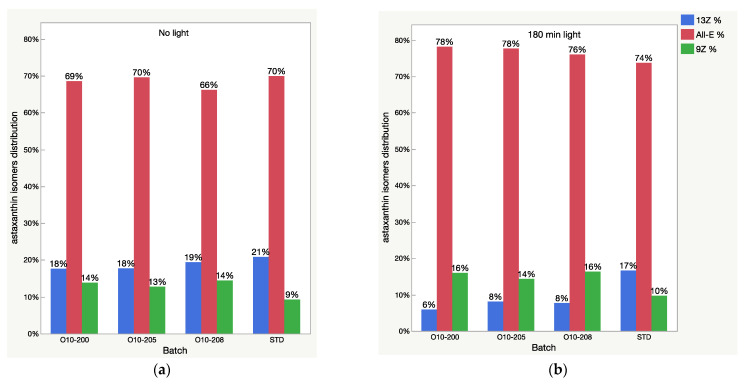
Relative content of astaxanthin isomers (isomeric distribution) in oleoresin samples before (**a**) and after 3 h (**b**) of light exposure.

**Figure 6 molecules-28-00847-f006:**
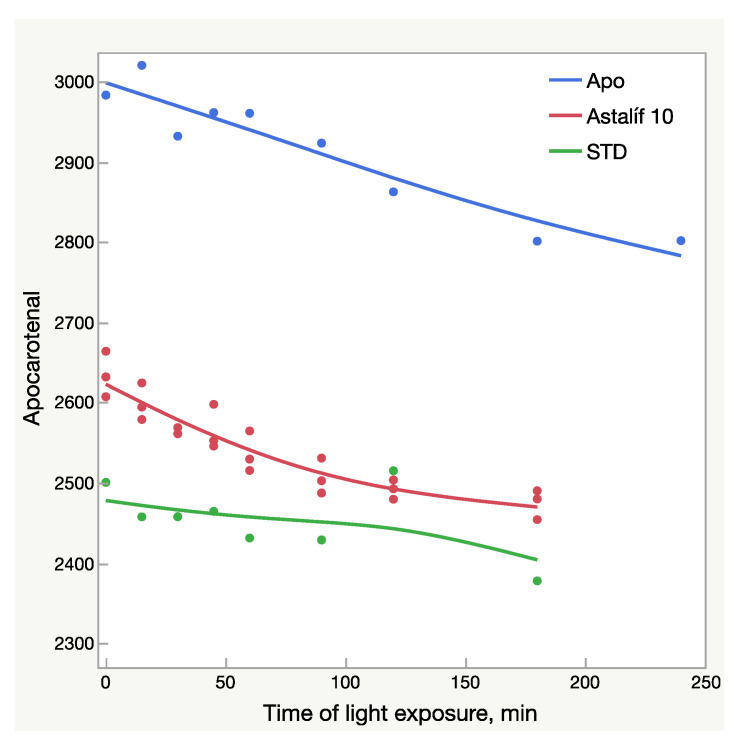
Light-dependent change of the apocarotenal peak area (mAU × s); AU—absorbance units; Apo—pure apocarotenal solution; Astalif 10—apocarotenal added to the samples O10-200, O10-205, and O10-208; STD—apocarotenal added to the reference standard solution.

**Figure 7 molecules-28-00847-f007:**
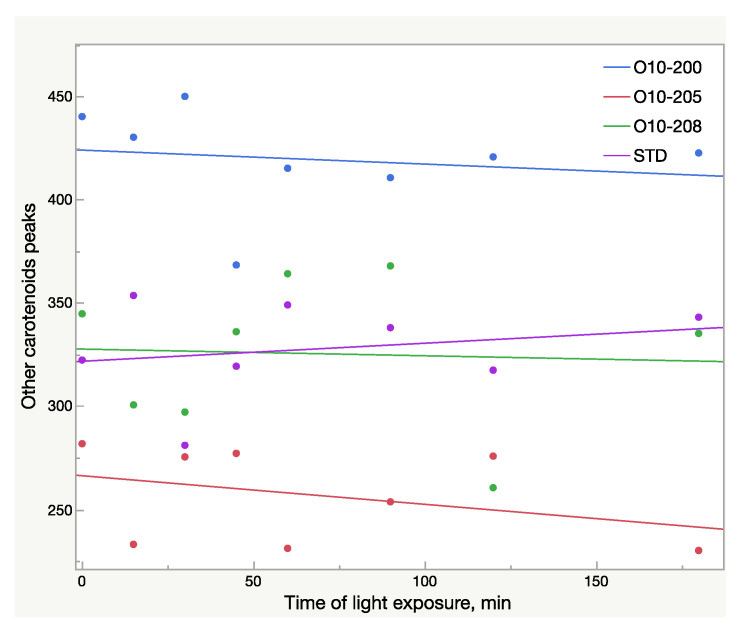
Total area (mAU × s) of other carotenoid peaks in oleoresin samples broken down by sample lots within 3 h of light exposure; AU—absorbance units.

**Figure 8 molecules-28-00847-f008:**
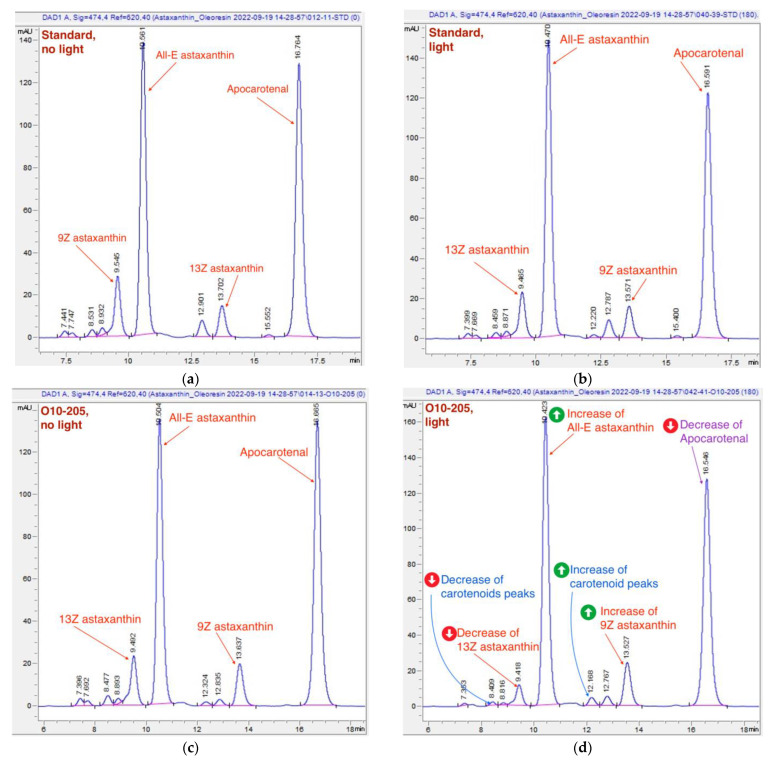
Typical chromatograms of oleoresin samples. (**a**) The reference standard chromatogram, “no-light” approach; (**b**) the reference standard chromatogram, 180 min of light exposure; (**c**) a chromatogram of O10-205, “no-light” approach; (**d**) a chromatogram of O10-205, 180 min of light exposure. The light-induced shifts are more pronounced on the chromatograph of batch O10-205 (**d**) than on the standard one (**b**): the change of astaxanthin, carotenoids and apocarotenal is marked in red, blue, and purple respectively.

**Figure 9 molecules-28-00847-f009:**
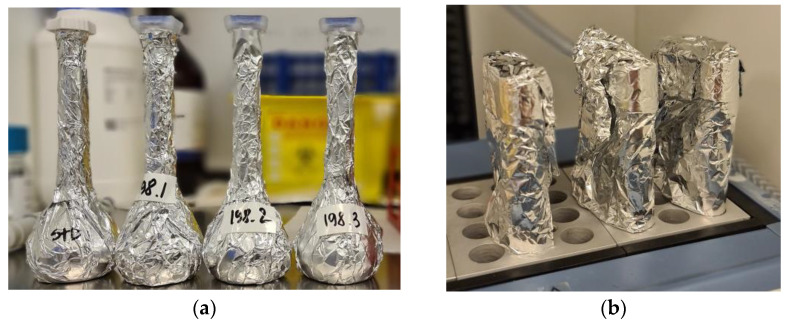
Preparation of oleoresin samples for the HPLC analysis. (**a**) 100-mL amber glass flasks wrapped with aluminium foil with stock solutions. (**b**) Pyrex tubes wrapped with aluminium foil during hydrolysis.

**Table 1 molecules-28-00847-t001:** The parameters of the logistic curve of the UV/Vis growth depending on time of light exposure.

Curve Parameters	Amber Glass Flask	Clear Glass Flask
a	0.007 *	0.017 **
b	−343.2 **	−139.6 **
c	0.1090 **	0.1090 **

* *p* < 0.05 indicates significant values, ** *p* < 0.001 indicates highly significant values.

**Table 2 molecules-28-00847-t002:** The parameters of the logistic curve of the UV/Vis growth depending on time of light exposure broken down by sample lots.

Curve Parameters	Amber Glass Flask				Clear Glass Flask			
	O10-200	O10-205	O10-208	Standard	O10-200	O10-205	O10-208	Standard
a	0.011 **	0.009 **	0.008 **	0.0003 ^&^	0.026 **	0.011 **	0.019 **	0.007 *
b	−222.3 **	−266.4 **	−302.4 **	6077.5 ^&^	−98.7 **	−183.5 **	−115.8 **	−322.0 **
c	0.1086 **	0.1080 **	0.1084 **	0.7760 ^&^	0.1084 **	0.1119 **	0.1107 **	0.1109 *

* *p* < 0.05 indicates significant values, ** *p* < 0.001 indicates highly significant values, ^&^ *p* > 0.95 indicates insignificant parameters.

**Table 3 molecules-28-00847-t003:** The parameters of the logistic curve of the astaxanthin level growth (HPLC) depending on time of light exposure broken down by sample lots.

Curve Parameters	Clear Glass Flask			
	O10-200	O10-205	O10-208	Standard
a	0.009 *	0.008 *	0.010 *	0.0003 ^&^
b	−169.1 *	−220.3 *	−162.3 *	7753.4 ^&^
c	0.1224 *	0.1169 *	0.1186 *	1.8671 ^&^

* *p* < 0.0001 indicates highly significant values and ^&^ *p* > 0.95 indicates insignificant parameters.

**Table 4 molecules-28-00847-t004:** Oleoresin lots and internal standard used for the study.

Lot	Main Components	UV, %	HPLC, %	UV/HPLC, %
USP reference astaxanthin standard, lot R120D0	Algal oleoresin ^1^	10.3	10.0 ^2^	103.0%
Sigma apocarotenal standard, lot BCCF4518	Trans-β-Apo-8′-carotenal	NA	97.7 ^2^	NA
O10-200	Algal oleoresin ^1^ (73%), Sunflower Oil (27%)	10.77	10.07 ^3^	107.0
O10-205	Algal oleoresin ^1^ (63%), Sunflower Oil (36%)	10.34	10.17 ^3^	101.7
O10-208	Algal oleoresin ^1^ (83%), Sunflower Oil (17%)	10.55	10.06 ^3^	104.9

^1^ Fatty acids and astaxanthin esters from *Haematococcus pluvialis*; ^2^ level indicated on the label; ^3^ analyzed during production by HPLC “no-light” approach according to the USP42 monograph.

**Table 5 molecules-28-00847-t005:** The HPLC sequence of analyzed oleoresin samples and apocarotenal ^1^.

Line No.	Sample Code	Description
1	Apo(0)	Apocarotenal sample, no-light
2	Apo(15)	Apocarotenal sample, 15 min light exposure
3	Apo(30)	Apocarotenal sample, 30 min light exposure
4	Apo(45)	Apocarotenal sample, 45 min light exposure
5	Apo(60)	Apocarotenal sample, 60 min light exposure
6	Apo(90)	Apocarotenal sample, 90 min light exposure
7	Apo(120)	Apocarotenal sample, 120 min light exposure
8	Apo(180)	Apocarotenal sample, 180 min light exposure
9	STD(0)	USP reference standard, no-light
10	O10-200(0)	Astalíf 10, O10-200, no-light
11	O10-205(0)	Astalíf 10, O10-205, no-light
12	O10-208(0)	Astalíf 10, O10-208, no-light
13	STD(15)	USP reference standard, 15 min light exposure
14	O10-200(15)	Astalíf 10, O10-200, 15 min light exposure
15	O10-205(15)	Astalíf 10, O10-205, 15 min light exposure
16	O10-208(15)	Astalíf 10, O10-208, 15 min light exposure
17	STD(30)	USP reference standard, 30 min light exposure
18	O10-200(30)	Astalíf 10, O10-200, 30 min light exposure
19	O10-205(30)	Astalíf 10, O10-205, 30 min light exposure
20	O10-208(30)	Astalíf 10, O10-208, 30 min light exposure
21	STD(45)	USP reference standard, 45 min light exposure
22	O10-200(45)	Astalíf 10, O10-200, 45 min light exposure
23	O10-205(45)	Astalíf 10, O10-205, 45 min light exposure
24	O10-208(45)	Astalíf 10, O10-208, 45 min light exposure
25	STD(60)	USP reference standard, 60 min light exposure
26	O10-200(60)	Astalíf 10, O10-200, 60 min light exposure
27	O10-205(60)	Astalíf 10, O10-205, 60 min light exposure
28	O10-208(60)	Astalíf 10, O10-208, 60 min light exposure
29	STD(90)	USP reference standard, 90 min light exposure
30	O10-200(90)	Astalíf 10, O10-200, 90 min light exposure
31	O10-205(90)	Astalíf 10, O10-205, 90 min light exposure
32	O10-208(90)	Astalíf 10, O10-208, 90 min light exposure
33	STD(120)	USP reference standard, 120 min light exposure
34	O10-200(120)	Astalíf 10, O10-200, 120 min light exposure
35	O10-205(120)	Astalíf 10, O10-205, 120 min light exposure
36	O10-208(120)	Astalíf 10, O10-208, 120 min light exposure
37	STD(180)	USP reference standard, 180 min light exposure
38	O10-200(180)	Astalíf 10, O10-200, 180 min light exposure
39	O10-205(180)	Astalíf 10, O10-205, 180 min light exposure
40	O10-208(180)	Astalíf 10, O10-208, 180 min light exposure

^1^ System suitability testing samples not shown.

**Table 6 molecules-28-00847-t006:** HPLC mobile phase gradient.

Time, min	Methanol, %	t-Butylmethylether, %	Phosphoric Acid, %
0	81	15	4
15	66	30	4
23	16	80	4
27	16	80	4
27.1	81	15	4
35	81	15	4

## Data Availability

Not applicable.

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
