# Peer review of "Light Increases Astaxanthin Absorbance in Acetone Solution through Isomerization Reactions"

_molecules, 2023, doi:10.3390/molecules28020847_

Round 1

Reviewer 1 Report

Molecules

REVIEW PAPER

Title of paper: Light Increases Astaxanthin Absorbance in Acetone Solution through Isomerization Reactions

Author/s: Oleksandr Virchenko, Tryggvi Stefánsson

Evaluation:

Manuscript is clear and concisely written. Abstract adequately describes the study, principle results and conclusions. Employed experimental methods are adequate, sufficiently clear and complete to allow repetition of the work. Data are properly analyzed and interpreted to support the conclusions. Relevant issues in discussion are adequately discussed.

Recommendation:

Accept with minor revision.

Comments:

Figure 8: The markings are not visible.

Materials and methods: divide the materials and methods into more paragraphs to make it more clear.

References: Can you include more references?

Author Response

Response to Reviewer 1 Comments

Thank you for all your comments.

Point 1: Figure 8: The markings are not visible.

Response 1: The images will be updated and rearranged to make them visible.

Point 2: Materials and methods: divide the materials and methods into more paragraphs to make it more clear.

Response 2: This will be done.

Point 3: References: Can you include more references?

Response 3: There is not much information regarding cis to trans transformation of astaxanthin, mainly the conversion of All-E form to 13Z form under heat exposure. Anyway, we will try to include more references to the relevant articles.

Reviewer 2 Report

This work demonstrates that light-inducing isomerization of astaxanthin is an important reason for carotenoid transformation. It disclosed the potential flaws in the USP method to measure Astaxanthin because of the different changes in isomeric distribution. The experimental design and the results are satisfactory. 

There is an issue that needs to be revised or supplemented:

Materials and Methods: The details of the UV/VIS and HPLC conditions should be described.

Author Response

Response to Reviewer 2 Comments

Point 1. Materials and Methods: The details of the UV/VIS and HPLC conditions should be described.

Response 1. Thank you for your review. The methods will be rewritten in detail to be full and explanatory allowing repetition of the work without reference to the USP monograph.